# *KRAS* Mutation Dictates the Cancer Immune Environment in Pancreatic Ductal Adenocarcinoma and Other Adenocarcinomas

**DOI:** 10.3390/cancers13102429

**Published:** 2021-05-18

**Authors:** Meichen Gu, Yanli Gao, Pengyu Chang

**Affiliations:** 1Department of Radiation Oncology & Therapy, The First Hospital of Jilin University, Changchun 130021, China; gumc20@mails.jlu.edu.cn; 2Department of Pediatric Ultrasound, The First Hospital of Jilin University, Changchun 130021, China; gaoyanli@jlu.edu.cn

**Keywords:** *KRAS* gene, pancreatic ductal adenocarcinoma, cancer immunity, immune checkpoint blockade

## Abstract

**Simple Summary:**

The vast majority of patients with pancreatic ductal adenocarcinomas harbor *KRAS* mutations in their tumors. Functionally, mutated *KRAS* is not only dedicated to tumor cell proliferation, survival and invasiveness, but also causing the immunosuppression in this cancer. In this situation, current data indicating the therapeutic effects of immune checkpoint inhibitors on pancreatic ductal adenocarcinomas are still not satisfying. In order to reflect the present bottleneck of immune checkpoint inhibitors in managing this cancer, we mainly provide information associated with the mechanism by which *KRAS* mutations establish the immunosuppressive milieus in pancreatic ductal adenocarcinomas. Together with other advances in this field, future directions to overcome the *KRAS* mutation-induced immunosuppression in pancreatic ductal adenocarcinomas are raised as well. Meanwhile, lung adenocarcinomas and colorectal adenocarcinomas are enumerated to compare with pancreatic ductal adenocarcinomas, aiming to indicate the specificity of *KRAS* mutations in dictating tumoral immune milieus among these cancers.

**Abstract:**

Generally, patients with pancreatic ductal adenocarcinoma, especially those with wide metastatic lesions, have a poor prognosis. Recently, a breakthrough in improving their survival has been achieved by using first-line chemotherapy, such as gemcitabine plus nab-paclitaxel or oxaliplatin plus irinotecan plus 5-fluorouracil plus calcium folinate. Unfortunately, regimens with high effectiveness are still absent in second- or later-line settings. In addition, although immunotherapy using checkpoint inhibitors definitively represents a novel method for metastatic cancers, monotherapy using checkpoint inhibitors is almost completely ineffective for pancreatic ductal adenocarcinomas largely due to the suppressive immune milieu in such tumors. Critically, the genomic alteration pattern is believed to impact cancer immune environment. Surprisingly, *KRAS* gene mutation is found in almost all pancreatic ductal adenocarcinomas. Moreover, *KRAS* mutation is indispensable for pancreatic carcinogenesis. On these bases, a relationship likely exists between this oncogene and immunosuppression in this cancer. During pancreatic carcinogenesis, *KRAS* mutation-driven events, such as metabolic reprogramming, cell autophagy, and persistent activation of the yes-associated protein pathway, converge to cause immune evasion. However, intriguingly, *KRAS* mutation can dictate a different immune environment in other types of adenocarcinoma, such as colorectal adenocarcinoma and lung adenocarcinoma. Overall, the *KRAS* mutation can drive an immunosuppression in pancreatic ductal adenocarcinomas or in colorectal carcinomas, but this mechanism is not true in *KRAS*-mutant lung adenocarcinomas, especially in the presence of *TP53* inactivation. As a result, the response of these adenocarcinomas to checkpoint inhibitors will vary.

## 1. Introduction

In humans, patients with pancreatic ductal adenocarcinomas (PDACs) commonly have a poor prognosis. As reported in 2018, the five-year survival rate of PDAC patients is only 9% [1]. The biology of PDAC is aggressive, and a certain portion of patients will die from disease-related complications rather than this disease itself [2]. Traditional approaches for managing this cancer include surgery, radiotherapy and chemotherapy. To exploit the genomic characteristics of PDAC, some molecular targeted approaches have been developed. These approaches have exhibited therapeutic effects in a small portion of metastatic cases carrying specific driver alterations, such as the treatment of cases with a germline breast cancer susceptibility gene 1 (*BRCA1*) or *BRCA2* mutation using olaparib, a poly ADP-ribose polymerase (PARP) inhibitor, or the treatment of cases with neuro trophin receptor kinase gene (*NTRK*) gene fusions using larotrectinib or entrectinib [3]. Recently, immune checkpoint blockade (ICB) therapy has opened a new era in the comprehensive treatment of cancers. In metastatic PDAC, only patients with the high microsatellite instability (MSI-H) or deficient mismatch repair (dMMR) phenotype in their tumors are reported to benefit from the ICB therapy with pembrolizumab, an anti-programmed death-1 (PD-1) drug [4]. However, the MSI-H/dMMR phenotype is rarely detected in PDAC. For those patients without the MSI-H/dMMR phenotype, available data indicate that their responses to monotherapy by using ICB drugs are extremely poor [5].

The existing immune environment in tumors will impact the effectiveness of ICB therapy [6]. In PDAC, the tumor milieu is generally immunosuppressive [7]. Recently, driver oncogenes have been recognized to play a convincing role in the cancer immune status [8]. In PDAC, the *Kirsten rat sarcoma* virus oncogene homolog (*KRAS*) gene is broadly mutated [9]. *KRAS* mutations in PDAC include those induced by a missense mutation in codon 12 or codon 13, leading to a replacement of the original glycine (G) by other amino acids, thus causing persistent activation of the *KRAS* protein in this setting [9]. The *KRAS* mutation acts as a driver to cause PDAC occurrence and progression together with the concomitant inactivation of other genes, such as tumor protein P53 gene (*TP53*), cyclin dependent kinase inhibitor 2A gene (*CDKN2A*) and SMAD family member 4 gene (*SMAD4*) [10,11] (Figure 1). In this process, the *KRAS* mutation will also lead to activation of downstream pathways that can improve cancer cell survival, proliferation, immune evasion and drug resistance [7,9]. Concerning immunosuppression in PDAC, the *KRAS* mutation utilizes several routes to achieve this goal, such as activating the yes-associated protein (YAP)- tafazzin (TAZ) pathway and its downstream Janus kinase-signal transducers and activators of transcription 3 (JAK-STAT3) signaling [12], inducing cell autophagy-associated major histocompatibility complex-I (MHC-I) degradation by reprogramming glucose metabolism [13,14], and synergizing with other genetic alterations (e.g., *TP53* inactivation) [15] (Figure 1). Consequently, PDAC tumors can be infiltrated by myeloid cells with pro-cancer functions, such as neutrophils, myeloid-derived suppressive cells (MDSCs) and M2-like macrophages [7].

In addition to PDAC, other cancers in humans, such as colorectal adenocarcinomas (CRACs) and lung adenocarcinomas (LUACs), also harbor a high prevalence of *KRAS* mutations [9]. Although *KRAS* mutation has been revealed to correlate with immune evasion in PDAC [13,14], the situation in LUAC appears to be different because LUAC tumors with *KRAS* mutation plus *TP53* inactivation commonly have massive infiltration of tumoricidal T cells and PD-L1 upregulation [16]. Moreover, clinical data support that LUAC patients with this pattern of tumor immunity can largely benefit from anti-PD-1 monotherapy [17]. Similarly, *KRAS* mutation is able to cause immunosuppression in CRAC tumors as well. However, unlike in PDAC, the published data suggest that CRAC patients with this phenotype can benefit from a combinational strategy featuring conventional therapy plus an ICB drug [18]. Importantly, despite having *KRAS* mutation, PDACs, CRACs and LUACs differ in their tumor immune status (Table 1).

Given the above information, this review will focus on the role of *KRAS* mutation in dictating pancreatic carcinogenesis and the cancer immune status in PDAC, aiming to illustrate the response of PDAC to ICB therapy in published data and to provide new insights into the use of ICB therapy in PDAC treatment. In addition, we will consider other *KRAS*-mutant cancers, such as CRAC and LUAC, and compare them with PDAC, aiming to uncover the mechanism by which *KRAS* mutation dictates the cancer immune status across these adenocarcinomas.

## 2. The Carcinogenic Role of *KRAS* Mutation in PDAC

Human PDAC exclusively has *KRAS* mutation rather than neuroblastoma RAS viral oncogene homolog gene (*NRAS*) or *Harvey rat sarcoma* viral oncogene homolog gene (*HRAS*) mutation [9]. Overall, 97.7% of PDAC cases are detected to have the *KRAS* mutation [9]. *G12D*, *G12V* and *G12R* are the three most common missense forms of *KRAS* mutation in PDAC, while the *G12D* missense mutation is the most frequent among them [9] (Table 1). Physiologically, the normal *KRAS* protein has GTPase activity, but these missense variants generate a KRAS protein that stably binds with GTP, thus constitutively activating MAPK and PI3K-Akt pathways, two classical pathways responsible for maintaining cell survival and proliferation [35] (Figure 1). For example, in mice bearing PDAC, the *KRAS^G12D^* mutation was revealed to activate the MAPK and PI3K-Akt pathways to increase the cellular content of Krüppel-like factor 5 (KLF5), which was required for PDAC cell proliferation [36]. Consistently, in human cell lines, MAPK activation upon *KRAS* mutation was revealed to induce posttranscriptional modification of YAP, and *KRAS* mutation was able to augment the transcriptional activity of YAP on its target genes [37]. Functionally, YAP-TAZ activation was demonstrated to be required for pancreatic carcinogenesis in mice carrying the *KRAS^G12D^* mutation: YAP and TAZ protein levels were upregulated in each stage of PDAC pathogenesis, including pancreatitis, acinar-to-ductal metaplasia (ADM) and pancreatic intraepithelial neoplasia (PanIN), and double knockout of *Yap* and *Taz* genes significantly mitigated *KRAS^G12D^* mutation-induced ADM and PanIN lesions [12]. In fact, YAP is essential for maintaining glucose metabolism in normal pancreatic epithelial cells [37]. This means that the *KRAS* mutation potentially induces a metabolic dysregulation of glucose. In a previous study, the *KRAS^G12D^* mutation was revealed to induce an upregulation of the gene encoding NIX, a critical protein for inducing mitophagy, thus restricting glucose flux into mitochondria (Figure 1). Via this mechanism, glucose metabolism in PDAC cells could be switched to favor glycolysis, and the antioxidant program could be activated, thus facilitating cell proliferation [13]. To understand the relationship among *KRAS* mutation, the antioxidant program and cell proliferation in PDAC, another study conducted by the same team reported that the *KRAS^G12D^* mutation could activate the nuclear-related factor 2 (Nrf2)-related antioxidant program in pancreatic epithelial cells of mice; in addition, PanIN cells from Nrf2-deficient mice were less proliferative than those without Nrf2 deficiency [38]. Consistent with this finding, inhibiting glutathione synthesis in PanIN cells without Nrf2 deficiency decreased their proliferation [38]. Collectively, these results show that *KRAS* mutation impacts the proliferation of PDAC cells in a metabolic manner (Figure 1).

## 3. The *KRAS* Mutation and Immune Environment in PDAC

In addition to impacting cell survival, proliferation and nutrient metabolism during pancreatic carcinogenesis, *KRAS* mutations also function in controlling the cancer immune environment. As documented, competition for glucose between cancer cells and stromal immune cells serves as a route for immune evasion of tumors [39]. As evidenced in mice, pancreatic epithelial cells carrying the *KRAS^G12D^* mutation and *Lkb1* inactivation were revealed to enhance their proliferation by overly consuming glucose [19]. In addition, in mice bearing pancreatitis-induced ADM, KLF5 deficiency was revealed to suppress STAT3 activation [36]. Generally, STAT3 activation correlates with immune suppression in cancers [40]. In the presence of *KRAS^G12D^* mutation, *Stat3* was revealed to be required for the development of ADM and PanIN during pancreatic carcinogenesis in mice [41]. In this model, IL-6 family cytokines were found to serve as inflammatory stimuli for STAT3 activation [41]. In another mechanism, *KRAS^G12D^* mutation-induced upregulation of YAP and TAZ was revealed to potently activate the downstream JAK-STAT3 pathway during pancreatic carcinogenesis in mice [12] (Figure 1). In fact, mutant *KRAS* can cooperate with extracellular stimuli, such as inflammation, the gut microbiota and gastrointestinal peptides, to persistently activate downstream YAP-TAZ signaling, which undermines immune surveillance against PDAC cells in addition to improving their proliferation, invasion, survival and metabolism [42]. In PDAC, a high expression of YAP was revealed to correlate with a poor histological grade of tumor cells [43], a high risk of metastasis and a poor prognosis of patients [44].

Mechanistically, *KRAS* mutation-induced activation of YAP enables PDAC cells to release IL-4, IL-6, IL-13, MCP-1 and CSF-1, which promote the recruitment of tumor-associated macrophages (TAMs) into tumors and induce them to proliferate and polarize into an M2-like phenotype [45] (Figure 1). In addition, the prevalence of *TP53* inactivation is only second to the prevalence of *KRAS* mutation in PDAC [10], meaning that a large portion of patients concomitantly harbor *KRAS* mutation and *TP53* inactivation [10]. To evaluate the function of this genetic alteration pattern in pancreatic carcinogenesis, concomitantly transgenic mutations of *KRAS^G12D^* and *Tp53^R172H^* were introduced into the pancreas of mice, resulting in PDAC formation and metastasis [46]. In this research, the *Tp53^R172H^* mutation was found to accelerate chromosomal instability in the presence of the *KRAS^G12D^* mutation compared with wild-type *Tp53* [46]. In addition, the *Tp53* inactivation cooperated with the *KRAS* mutation to induce PDAC cells to secrete chemokine C-X-C motif receptor (CXCR3)/chemokine C-C motif receptor (CCR2)-associated chemokines and CSF-1, thus recruiting myeloid-derived suppressive cells (MDSCs) into PDAC tumors and promoting the expansion of MDSCs [15]. In addition, PDAC tumors with *KRAS* mutation plus *Tp53* inactivation had increased numbers of Treg cells compared with PDAC tumors with only *KRAS* mutation [15]. In tumors with both alterations, the Treg cells presented upregulation of CD25, glucocorticoid-induced tumor necrosis factor receptor (GITR) and killer cell lectin like receptor G1 (KLRG1), indicating increased suppressive ability [15]. In addition, Th1 and CD8^+^ T cell-mediated anticancer responses were attenuated [15]. Conversely, pancreas-specific knockout of *Yap* in mice carrying *KRAS^G12D^/Tp53^R172H^* co-mutation restored the expression of cytotoxicity-associated genes by CD8^+^ T cells in addition to preventing MDSC accumulation [47]. This result suggests that *Yap* is required for *KRAS* mutation-induced immunosuppression in PDAC tumors.

In concert with the *KRAS* mutation, alterations in environmental, genetic and molecular levels, such as hypoxia, *LKB1* mutation, *PTEN* loss, PIK3CA activation, WNT/β-catenin activation, FAK activation and *MYC* proto-oncogene (MYC) activation also contribute to immune suppression in PDAC tumors [29,30] (Figure 1). For example, hypoxia can activate HIF-1α, and moreover, HIF-1α activation is potent in inducing tumoral angiogenesis by increasing the expression of VEGF [48]. This event also occurs in PDAC [49]. As documented, VEGF is a potent cytokine that undermines anticancer immunity by dictating the expansion, phenotypic conversion and suppressive function of tumor-infiltrating immune cells, such as MDSCs, TAMs, dendritic cells (DCs) and Treg cells [48] (Figure 1). In response to hypoxia, some infiltrating immune cells, such as DCs and TAMs, and the endothelium can induce self-expression of PD-L1 molecule, thus impairing the infiltration, survival and effector function of tumoricidal T cells [48]. In addition to immune cells, tumor cells are critical sources of PD-L1. For example, the transcriptional activation of MYC enables PDAC cells to upregulate PD-L1 expression [50]. In addition, mixed lineage leukemia protein-1 (MLL1) can upregulate PD-L1 expression: as a histone methyltransferase, MLL1 can accelerate histone 3 lysine 4 (H3K4) trimethylation in the promoter of the gene encoding PD-L1 [51]. Via these actions, immune evasion in PDAC tumors can be facilitated. Thus, as documented, features of the immune milieu in PDAC tumors include infiltration of cancer-supportive cells (e.g., cancer-associated fibroblasts (CAFs), Treg cells, suppressive neutrophils, indoleamine 2,3-dioxygenase (IDO)-producing DCs, M2-like TAMs and MDSCs), upregulation of suppressive cytokines (e.g., nitric oxide, hyaluronic acid, IL-6, IL-10, VEGF, TGF-β, CSF-1, GM-CSF, CXCL1, CXCL8, CXCL12 and CXCL13), angiogenesis and ‘T cell exclusion’ [7,52,53] (Figure 1). In fact, both in humans and mice, although PDAC tumors were found to harbor tumoricidal T cell infiltrates, few of them were found in the vicinity of PDAC cells, a phenomenon known as ‘T cell exclusion’ [53,54] (Figure 1). This exclusion is a critical mechanism by which intratumoral cells, such as CAFs, M2-like TAMs and MDSCs, encourage PDAC cells to escape T cell attack [52]. In support of this mechanism, CAF-derived CXCL12 was demonstrated to show a high affinity to PDAC cells, whereas inhibition of CXCR4 by using AMD3100 could significantly limit the tumor growth of mice bearing PDAC in a T cell-dependent manner [54]. Moreover, upon CXCR4 inhibition, PDAC cells could be besieged by massive numbers of T cells [54]. In addition, myeloid-derived Ly6G^Low+^/F4/80^+^ macrophages served as extratumoral cells that caused T cell exclusion from the PDAC tumors of mice [55]. In summary, due to the lack of tumoricidal T cells and the enrichment of immunosuppressive cells and cytokines, the immune milieu of PDAC tumors is generally cancer-supportive (Figure 1).

## 4. Current Status of Immune Checkpoint Blockade Therapy for PDAC

Since the tumoral milieu of PDAC is immunosuppressive, ICB therapy is anticipated to have low effectiveness in this cancer. In fact, several lines of clinical data have confirmed this speculation, and the effectiveness of monotherapy by using ICB drugs in patients with metastatic PDAC remains disappointing [5]. For example, a phase II study reported that as a second- or later-line therapy for metastatic PDAC, durvalumab (an anti-PD-L1 drug) alone and durvalumab plus tremelimumab (an anti-cytotoxic T lymphocyte-associated antigen-4 (CTLA-4) drug) had objective response rate (ORR) values of 0% and 3.1%, respectively [56] (Table 2). Prior to this study, in order to improve the effectiveness of ICB therapy, a phase I study employed stereotactic body radiation therapy (SBRT) in combination with ICB therapy (in this case pembrolizumab) to upregulate the expression of the genes encoding PD-L1 and MHC-I in tumor cells and improve the production of tumor-associated antigens (TAAs) by recruiting tumoricidal T cells and by improving the production of IFN-γ by CD8^+^ T cells [18] as a strategy against metastatic cancers, and this combination achieved an ORR of 13.2% among enrolled patients [57]. However, this study only included three patients with metastatic PDAC, and their ORR to this strategy was not reported. Recently, a single-center phase I study tested SBRT plus durvalumab with or without tremelimumab as a second- or later-line therapy for metastatic PDAC patients [58]. Unexpectedly, the ORR for this strategy was only 5.1% [57]. As with radiotherapy, chemotherapy agents exert cytotoxicity to induce immunogenic cell death (ICD) as well [59]. To evaluate the synergistic effect of chemotherapy plus ICB therapy, a phase I study was carried out, and 2 of 11 patients with metastatic PDAC achieved a partial response after receiving gemcitabine-based chemotherapy plus pembrolizumab [60]. Yet, these two patients were chemotherapy-naïve. In contrast, the remaining patients had received at least one line of chemotherapy before receiving this therapy combination, which had produced stable disease in most of them [60]. Consistent with this finding, another phase I study concluded that an anti-CTLA-4 drug (ipilimumab) plus gemcitabine exhibited no advantages over gemcitabine alone in increasing the ORR of patients with metastatic PDAC [61]. Notably, most patients had received at least one line therapy prior to being enrolled in the study. Hence, the above data suggest that ICB drugs are not effective in significantly shrinking the size of PDAC tumors when used as a second- or later-line therapy regardless of whether they are used alone or in combination with radiotherapy or chemotherapy (Table 2).

In fact, metastatic cancers commonly show clonal evolution of tumor cells as the therapies are engaged [67], and this scenario is suitable for ICB therapy [68]. As reported, first-line chemotherapy using [FOLFIRINOX] (oxaliplatin plus irinotecan plus 5-fluorouracil plus calcium folinate) [69] or [GA] (gemcitabine plus nab-paclitaxel) [70] regimens significantly prolonged the overall survival of patients with metastatic PDAC compared with gemcitabine monotherapy, implying that these combination regimens are more effective in killing tumor cells. In this regard, adding ICB drugs to intensive chemotherapy is speculated to further improve the prognosis of patients, mainly because the increased burden of neoantigens derived from lysed cancer cells can potentially improve anticancer immunity when these antigens are successfully presented by DCs to peripheral T cells [71]. When such T cells migrate into the tumor, they can recognize the cancer clones sharing the neoantigens and then kill these cancer cells [71]. Supporting this theory, recent data from several phase I and II trials indeed revealed that as a first-line therapy, chemotherapy plus ICB therapy had improved effectiveness compared with as a second- or later-line therapy in metastatic PDAC (Table 2). For example, gemcitabine plus tremelimumab achieved an ORR of 10.5% [62]. The [GA] regimen plus nivolumab (an anti-PD-1 drug) or plus pembrolizumab achieved ORRs ranging from 18% to 50% [63,64]. More strikingly, when [GA] regimen was combined with durvalumab plus tremelimumab, the ORR was 73% [65]. In addition, an ORR of 80% was achieved when nivolumab was added to the regimen containing nab-paclitaxel, cisplatin, gemcitabine and paricalcitol [66]. These combinational strategies were tolerated by most enrolled patients. Therefore, although these trials had low patient numbers, their data at least provide new insights into the future management of metastatic PDAC by using chemotherapy plus ICB therapy in the first-line setting. Nevertheless, the prognostic value of this strategy in metastatic PDAC remains to be elucidated via randomized phase III trials.

Overall, the currently published data indicate an extremely low effectiveness of monotherapy by using ICI drugs or their combination with other conventional approaches, such as radiotherapy or chemotherapy, as second- or later-line therapies for metastatic PDAC (Table 2). In PDAC, only the MSI-H/dMMR phenotype is indicative of response to pembrolizumab. However, the prevalence of the MSI-H/dMMR phenotype in PDAC has been reported to be only 1~2% [25], but intriguingly, the MSI-H/dMMR phenotype was found to be strongly correlated with a high tumor mutational burden (TMB) and a wild-type KRAS and p53 molecular background [66]. Consequently, to achieve a breakthrough in the management of PDAC with *KRAS* mutation, a focus should be placed on eliminating the tumor cell- or stromal cell-induced barriers that counteract anticancer immunity. Recent studies in this field have revealed several strategies, such as adding an antiangiogenic drug [72], an anti-IL-6 antibody [73], an ataxia telaniectasia-mutated gene-coded protein (ATM) inhibitor [74], a CD40 agonist [55], a CSF1R inhibitor [75], a YAP inhibitor plus a pan-*RAF* proto-oncogene (RAF) inhibitor [43], a CXCR4 inhibitor [54], a PARP inhibitor [76], a *Listeria* vaccine plus an anti-CD25 antibody [77], a FAK inhibitor [78], a CCR2 inhibitor [79], an IDO inhibitor plus the GM-CSF-conjugated whole-cell PDAC vaccine (GVAX) [80], or the combination of anti-PD-1 and anti-PD-L1 monoclonal antibodies [81], that have been confirmed to improve the immune milieu and the effectiveness of ICB therapy in preclinical models of PDAC. On these bases, some strategies using ICB therapy plus GVAX or other means, such as CXCR4 inhibition, CSF1R inhibition and CD40 blockade, have been designed to treat PDAC patients in clinical trials [52]. A few strategies have exhibited their effectiveness, such as the success of GVAX plus ipilimumab in prolonging the survival of PDAC patients [82].

A few small molecular compounds, such as AMG510, MRTX849, ARS-3248/JNJ-74699157 or LY3499446, have been designed to antagonize cancer cells carrying the *KRAS^G12C^* mutation [83]. Among them, the data of AMG510 and MRTX849 are encouraging. For example, both in *KRAS^G12C^*-mutated LUAC and CRAC models, basic experiments revealed the tumoricidal activity of AMG510 or MRTX849 both in vitro and in vivo [84,85]; Likewise, administration of AMG510 or MRTX849 was confirmed to cause a significant shrinkage of tumors among patients with the *KRAS^G12C^*-mutated LUAC, CRAC or PDAC [84,85,86]. Moreover, the tumoral immune milieu can be improved by using such KRAS^G12C^ inhibitors. In the model of mice bearing *KRAS^G12C^*-muated CT-26 cell line-derived tumors, following AMG510 administration, T cells were found to significantly infiltrate into tumors [84]. Particularly, most of them were positive for CD8, and they presented a proliferating status upon AMG510 administration [84]. Mechanically, AMG510 administration could induce the upregulation of CXCL10 and CXCL11 by tumor cells, two crucial chemoattractant of T cells, thus causing an increasement of T cells in xenografted tumors [84]. Meanwhile, DCs including CD103^+^ cross-presenting pool and macrophages were found to increase their infiltration in xenografted tumors as well [84]. Functionally, CD103^+^ DCs are crucial for T cell priming and activation, while activated CD8^+^ T cells can produce IFN-γ, which enables tumor cells to increase their expression of MHC-I [84]. Thus, following AMG510 administration, the tumoral immune milieu was characterized by increased interferon signaling, antigen processing, chemokine production, cytotoxic activity and innate immune system stimulation [84]. Similar to AMG510, in the model of mice bearing *KRAS^G12C^*-muated CT-26 cell line-derived tumors, MRTX849 administration was revealed to induce the polarization of TAMs from M2 to M1, the infiltration of DCs, B cells and tumoricidal T cells in tumors, as well as the reduction of MDSCs in tumors [87]. Therefore, either AMG510 or MRTX849 plus an anti-PD-1 antibody were demonstrated to cause a durable shrinkage of xenografted tumors with *KRAS^G12C^* mutation [84,87]. In fact, data associated with the potential of AMG510 or MRTX849 in shifting tumoral immune milieu from a suppressive to a tumoricidal state are mainly collected from the model of CRAC, rather than PDAC [84,87]. In this regard, more efforts should be paid in the future to reveal whether *KRAS^G12C^* inhibition can improve the tumoral immune milieu of PDAC, thus enabling the combination of *KRAS^G12C^* inhibition and anti-PD-1 therapy to overcome the immunosuppression in PDAC. However, the frequency of *KRAS^G12C^* mutation only accounts for less than 3% among all PDAC cases, whereas approximately 50% of PDAC cases have the missense form of *G12D* [9]. In fact, adaptive transfer of CD8^+^ T cells that react with *KRAS^G12D^*-mutated tumor cells were demonstrated to be an effective approach in treating CRAC [88]. In order to benefit the majority of PDAC patients, drugs or new treatment strategies that target *G12D* missense mutation should deserve attention; in this scenario, the tumoricidal activity of newly developed approaches along with their potentials in improving tumoral immune milieu should be explored in the future.

## 5. Value of *KRAS* Mutation for Predicting Cancer Immune Status in Other Adenocarcinomas

As mentioned above, PDAC, CRAC and LUAC are the top three cancers harboring a high prevalence of *KRAS* mutations [9] (Table 1). In CRAC, the prevalence of *KRAS* mutation is 44.7% [9]. As in PDAC, *G12D* is the most frequent missense mutation that causes consecutive activation of *KRAS* protein in CRAC [9] (Table 1). Among *KRAS*-mutant CRAC cases, 35% to 50% of them are reported to have concomitant inactivation in APC and p53 [22]. In mice bearing CRAC, the *KRAS^G12D^* mutation was revealed to significantly increase the invasion and metastasis of cancer cells because conditional codeletion of *Apc* and *Tp53* concomitant with *KRAS^G12D^* mutation enabled primary and metastatic tumors to significantly upregulate the expression of the gene encoding TGF-β1, both a critical immunosuppressive cytokine [32] and a critical ligand of TGF-β/SMAD signaling that can dictate epithelial–mesenchymal transition (EMT) in CRAC cells [31]. Moreover, compared with patients with the wild-type *RAS*, CRAC patients harboring *KRAS* mutation generally have a poor prognosis [89].

However, the prevalence of the MSI-H/dMMR phenotype in CRAC is higher than that in PDAC. According to published data, the incidence of the MSI-H/dMMR phenotype in CRAC is 14% [26]. Currently, ICB therapy with pembrolizumab is recommended as the first-line therapy for metastatic CRAC with the MSI-H/dMMR phenotype, which has been confirmed as a reliable biomarker for predicting the outcome of ICB therapy by several lines of trial data [4,90,91]. Regardless, not all patients with this phenotype benefit from the ICB therapy [90,91]. In the KEYNOTE-177 study, chemotherapy plus bevacizumab was still more effective than pembrolizumab monotherapy in prolonging the progression-free survival of patients with metastatic disease, the MSI-H/dMMR phenotype and *KRAS* mutation [27]. Conversely, those patients without *KRAS* mutation did benefit more from pembrolizumab than chemotherapy plus bevacizumab [27]. Hence, these results suggest that *KRAS* mutation can undermine the effectiveness of pembrolizumab even in the presence of the MSI-H/dMMR phenotype. Critically, *KRAS* mutation was revealed to be enriched in CRAC with the microsatellite stability (MSS) or proficient mismatch repair (pMMR) phenotype [26]. However, published data reveal that patients with CRAC with the MSS/pMMR phenotype respond poorly to ICB therapy alone [28].

Similar to its role in PDAC (Table 1), *KRAS* mutation in CRAC with the MSS/pMMR phenotype generally correlates with immune suppression in the tumor. To address this issue, a study evaluated the role of *KRAS* mutation in dictating the cancer immune status of CRAC tumors [24], which were mainly classified into four subgroups, namely, consensus molecular subtype 1 (CMS1) (immune type), CMS2 (classical type), CMS3 (metabolic type) and CMS4 (mesenchymal type), according to which molecular pathways were enriched [26]. The results indicated that CMS2 or CMS3 tumors with *KRAS* mutation had a significantly reduced number of tumoricidal T cells compared with those without wild-type *KRAS* [24]. To explore the mechanism, experiments were performed in mice bearing CRAC with the *KRAS^G12D^* mutation plus conditional depletion of *Apc* and *Tp53*, and this genetic alteration pattern was found to enable the tumors to have increased numbers of MDSCs but decreased numbers of CD4^+^ or CD8^+^ T cells compared with the pattern of conditional codeletion of *Apc* and *Tp53* [33]. In detail, the *KRAS^G12D^* mutation was able to activate ERK, which showed a negative relationship with the expression of the gene encoding interferon-related factor 2 (IRF2) by tumor cells [24]. In return, *IRF2* inactivation upregulated the expression of the gene encoding CXCL3, a chemokine that attracts MDSCs into tumors, thus impairing the expansion and IFN-γ-producing function of tumoricidal T cells [33]. Notably, *KRAS* mutation-related IRF2 inactivation was revealed to correlate with a poor response of CRAC patients to ICB therapy [33]. Conversely, in mice bearing CRAC with the *KRAS^G12D^* mutation plus codeletion of *Apc* and *Tp53*, blocking CXCR2 on MDSCs improved the efficacy of anti-PD-1 therapy by increasing the number of CD8^+^ T cells but decreasing the number of Treg cells in tumors [33]. In fact, the CRAC tumors in these mice were revealed to resemble the CMS4 tumors in terms of some molecular signatures, such as the TGF-β/EMT signature [31]. As reported, the patients in the CMS4 subgroup commonly presented with rapid disease progression along with a poorer prognosis than the patients in other subgroups [26]. As such, *KRAS* mutation-induced immunosuppression potentially contributes to this process. In addition, CD8^+^ T cells that recognize the cancer cell clones carrying the *KRAS^G12D^* mutation have been shown to exist in human CRAC tumors [92]. To our knowledge, the recognition of tumor antigens by tumoricidal T cells is as critical as having these cells infiltrate into tumors. Therefore, ICB therapy is speculated to improve the anticancer effect of T cells on *KRAS*-mutant CRAC.

In fact, *KRAS*-mutant CRAC still has several differences from PDAC in tumor biology (Table 1). As mentioned above, CRAC patients with the MSS/pMMR phenotype appear to be inherently refractory to ICB therapy [28]. Unlike in PDAC, the data from clinical trials, such as VOLTAGE (chemoradiation followed by five doses of nivolumab before radical surgery as a neoadjuvant therapy for locally advanced rectal cancer) [93], MEDETREME (FOLFOX regimen plus durvalumab and tremelimumab as a first-line therapy for metastatic CRAC) [94] and REGONIVO (regorafenib plus nivolumab as a third-line therapy for refractory CRAC) [95], have confirmed that patients with MSS/pMMR tumors could benefit from ICB therapy-based combinational strategies. Certainly, a portion of patients harboring *KRAS* mutations in their tumors are included in these studies, thus helping to elucidate the role of chemoradiation, duplet chemotherapy or molecule-targeted therapy in boosting the tumoricidal milieu. In addition to using conventional means, several new means have been developed. As documented, *KRAS* mutation-driven molecular alterations cause CMS3 tumor cells to have dysregulated glucose, glutamine, fatty acid and lipid metabolism [26,34]. Targeting the metabolic abnormalities or blocking the downstream pathways affected by *KRAS* mutation, such as the MAPK and HIF-1-related pathways, has been shown to induce cancer cell death, potentially increasing the release of tumor antigens [34]. However, intriguingly, although *KRAS*-mutant CRAC cells have been revealed to consume glucose for their expansion, they are more resistant to glucose restriction than cells with wild-type *KRAS* [20]. This is another difference from PDAC cells, and murine pancreatic epithelial cells with the *KRAS^G12D^* mutation with *Lkb1* inactivation have been found to be sensitive to acute glucose restriction or glycolysis inhibition [39]. Consistent with this finding, LUAC cells in mice with homozygous *KRAS^G12D/G12D^* mutation were more sensitive to glucose restriction than those with heterogeneous *KRAS^G12D/wt^* mutation or *KRAS^wt/wt^*, and a higher consumption of glucose occurred in LUAC cells with the *KRAS^G12D/G12D^* mutation than in LUAC cells with other versions of *KRAS* [21]. Notably, *G12C* is the most common missense causing *KRAS* mutation in LUAC, with a prevalence of 30.9% in Western patients [9]. Nevertheless, *KRAS* mutation should not be regarded as a marker indicating immunosuppression in LUAC tumors because the immune milieu in *KRAS*-mutant LUAC tumors is heterogeneous. For example, the *KRAS*/*LKB1* and *KRAS*/*TP53* co-mutations enable LUAC patients to have dramatically different responses to ICB therapy because these two mutational patterns generally create a unique immune milieu in tumors (see details in [23]). Collectively, *KRAS* mutation can affect the cancer immune state in PDAC, CRAC and LUAC in different ways and contextures (Table 1).

## 6. Conclusions

The tumor milieu in PDAC is profoundly immunosuppressive, which renders monotherapy by using ICB drugs almost completely ineffective. Regarding the development of immunosuppression in PDAC, multiple factors are involved. Herein, *KRAS* mutation has been shown to be central in this process, because *KRAS* mutation can activate YAP-TAZ and JAK-STAT3 to elicit an immunosuppressive response, and this initial signaling can then be strengthened by coordination with *TP53* inactivation and other genetic or molecular alterations. Overall, *KRAS* mutation generally correlates with tumor immunosuppression in PDAC. Nevertheless, in CRAC and LUAC, *KRAS* mutation can dictate the cancer immune environment in different ways. In these cancers, the immune milieu varies despite the commonality of *KRAS* mutation. This notion can be exemplified by *KRAS*-mutant LUAC, which exhibits a varied response to ICB therapy depending on the types of genetic alterations that cooccur with the *KRAS* mutation.

## Figures and Tables

**Figure 1 cancers-13-02429-f001:**
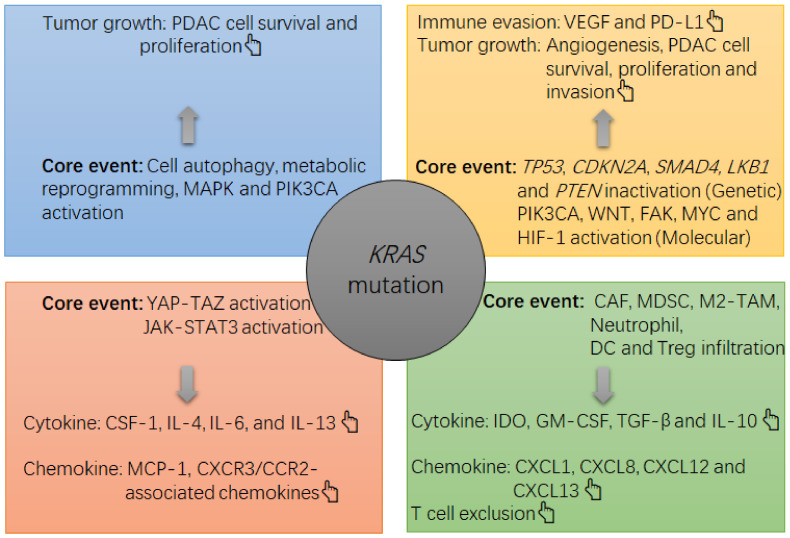
The note chart of *KRAS* mutation-induced growth and immunosuppression in PDAC tumors. The *KRAS* mutation causes a suppressive milieu in PDAC tumors mainly via the following routes, such as activation of mitogen-activated protein kinase (MAPK) and phosphatidylinositol 3-kinase (PI3K)-Akt, activation of YAP-TAZ and JAK-STAT3, and induction of cell autophagy and metabolic reprogramming in PDAC cells. In this context, the survival and proliferation of PDAC cells will be accelerated, and an overgrowth of tumor cells can cause a hypoxia within the tumor, which then activates hypoxia-induced factor 1 (HIF-1)α to upregulate the expression of gene encoding vascular endothelial growth factor (VEGF) by PDAC cells. VEGF is a potent cytokine that induces angiogenesis and immune evasion (e.g., programmed death ligand 1 (PD-L1) upregulation and tumoricidal T cell exhaustion). Meanwhile, PDAC cells can increase their production of suppressive cytokines and chemokines, such as interleukin 4 (IL-4), IL-6, IL-13, macrophage-colony stimulating factor 1 (CSF-1) and monocyte chemotactic protein 1 (MCP-1), which then recruit and increase the survival and suppressive function of immune infiltrates including cancer-associated fibroblasts (CAFs), MDSCs, M2-like tumor-associated macrophages (TAMs), indoleamine 2, 3-dioxygenase (IDO)-producing dendritic cells (DCs) and regulatory T cells (Treg cells). In this context, an overload of suppressive cells will increase the local levels of suppressive cytokines and chemokines, such as transforming growth factor-beta (TGF-β), IDO, IL-10, granulocyte-macrophage colony stimulating factor (GM-CSF), chemokine C-X-C motif ligand 1 (CXCL1), CXCL8, CXCL12 and CXCL13, thus strengthening the immunosuppression in the tumor (e.g., tumoricidal T cell exclusion). In concert with the *KRAS* mutation, other alterations at genetic and molecular levels, such as liver kinase B1 gene (*LKB1*) inactivation, *TP53* inactivation, phosphatase and tensin homolog gene (*PTEN*) inactivation, focal adhesion kinase (FAK) activation, phosphatidylinositol-4, 5-bisphosphate 3-kinase catalytic subunit alpha (PIK3CA) activation or Wingless/Integrated (WNT) activation, also contribute to the tumor growth (e.g., PDAC cell survival, proliferation and invasion) and immune evasion (PD-L1 upregulation).

**Table 1 cancers-13-02429-t001:** The comparison of immune-related characteristics among *KRAS*-mutant adenocarcinomas.

	Cancer	PDAC	CRAC	LUAC
Characters [Ref.]				
Prevalence of *KRAS* mutation		97.7% [9]	44.7% [9]	30.9% [9]
Hottest missense mutation in *KRAS*		*G12D* [9]	*G12D* [9]	*G12C* [9]
Sensitive to glucose restriction vs. *KRAS^wt^*		Yes [19]	Yes [20]	No [21]
Common alteration with *KRAS*		*TP53* inactivation [10]	*TP53* and *APC* inactivation [22]	*TP53* or *LKB1* inactivation [23]
General milieu of *KRAS*-mutant tumors		Immune-cold [7]	Immune-cold [24]	*KRAS*-only: immune-cold or hot [23] *TP53* inactivation: immune-hot [23] *LKB1* inactivation: immune-cold [23]
Number/function of tumoricidal T cells in *KRAS*-mutant tumors		Decrease/Decrease [7]	Decrease/Decrease [24]	*KRAS*-only: slight increase/decrease [23] *TP53* inactivation: significant increase/decrease [23] *LKB1* inactivation: significant decrease/decrease [23]
Major type of immune infiltrates in *KRAS*-mutant tumors		Myeloid suppressive cell [7]	Myeloid suppressive cell [24]	*KRAS*-only: T cell, macrophage, neutrophil [23] *TP53* inactivation: CD8^+^ T cell, CD45RO^+^ T cell [23] *LKB1* inactivation: myeloid suppressive cell [23]
Common presentation of the ICB therapy biomarker if *KRAS* mutation		pMMR/MSS [25]	pMMR/MSS [26]	*KRAS*-only: PD-L1 expression ↑ [23] *TP53* inactivation: PD-L1 expression ↑↑ [23] *LKB1* inactivation: PD-L1 expression ↓↓ [23]
Biomarker associated with the effectiveness of ICB therapy		dMMR/MSI-H [4]	dMMR/MSI-H [27]	PD-L1 [23]
Prevalence of dMMR/MSI-H in all cases		1~2% [25]	14% [26]	*NM*
Prevalence of positive expression of PD-L1 by tumor cells		*NM*	*NM*	Among *KRAS*-only tumors: 37.5% [23] Among *TP53* inactivation tumors: 68.8% [23] Among *LKB1* inactivation tumors: 10% [23]
General response to monotherapy using ICB drugs		Poor [5]	Poor [28]	*KRAS*-only tumor: Fair [23] *TP53* inactivation tumor: Excellent [23] *LKB1* inactivation tumor: Poor [23]
Core molecular events associated with *KRAS* mutation-induced immunosuppression		1. YAP-TAZ activation [12]; 2. JAK-STAT3 activation [12]; 3. Metabolic reprogramming of glucose and cell autophagy [13,14]; 4. In concert with other events, *TP53* inactivation [15], *LKB1* mutation [29,30], *PTEN* loss [29,30], WNT/β-catenin activation [29,30], FAK activation [29,30], PIK3CA activation [29,30] and MYC activation [29,30];	1. In concert with *APC* and *TP53* inactivation: TGF-β1 upregulation and EMT [31]; 2. TGF-β-induced immune suppression [32]; 3. *IRF2* inactivation [24,33]; 4. Metabolic dysregulation in glucose, glutamine, fatty acid and lipid [26,34]; 5. MAPK and HIF-1-related cascade activation [34];	1. ERK activation-induced PD-L1 upregulation [23] 2. Metabolic reprogramming of glucose [21] 3. In concert with *LKB1* inactivation: strengthening metabolic reprogramming of glucose and JAK-STAT3 activation [23]

PDAC: pancreatic ductal adenocarcinoma; CRAC: colorectal adenocarcinoma; EMT: epithelial-mesenchymal transition; LUAC: lung adenocarcinoma; APC: adenomatous polyposis coli protein; pMMR: proficient mismatch repair; MAPK: mitogen-activated protein kinase; MSS: microsatellite stability; dMMR: deficient mismatch repair; MSI-H: high microsatellite instability; ICB: immune checkpoint blockade; *NM*: no mention; PD-L1: programmed death-ligand 1; *TP53*: tumor protein P53 gene; *LKB1*: liver kinase B1 gene.

**Table 2 cancers-13-02429-t002:** The effectiveness of ICB therapy on PDAC.

Author [Ref.]	Year	Phase	Patient No.	ICB Drug	Other Treatment	ORR
• First-line therapy						
Aglietta M, et al. [62]	2014	I	34	Tremelimumab	Gemcitabine	10.5%
Wainberg ZA, et al. [63]	2019	I	50	Nivolumab	Gemcitabine + Nab- paclitaxel	18%
Wainberg ZA, et al. [64]	2017	I	17	Nivolumab	Gemcitabine + Nab- paclitaxel	50%
Renouf, et al. [65]	2018	II	11	Durvalumab + Tremelimumab	Gemcitabine + Nab-paclitaxel	73%
Borazanci, et al. [66]	2018	II	11	Nivolumab	Gemcitabine + Nab-paclitaxel + Cisplatin + Paricalcitol	80%
• Second- or later-line therapy						
Luke JJ, et al. [57]	2018	I	3	Pembrolizumab	SBRT: 30–50 Gy for 2–4 metastatic lesions	NR
O’Reilly EM, et al. [56]	2019	II	Arm A: 32 Arm B: 32	Durvalumab Durvalumab + Tremelimumab	No	0% 3.1%
Xie C, et al. [58]	2020	I	Arm A1: 14 Arm A2: 10 Arm B1: 19 Arm B2: 16	Durvalumab Durvalumab Durvalumab + Tremelimumab Durvalumab + Tremelimumab	SBRT: 8 Gy/1 fraction SBRT: 25 Gy/5 fractions SBRT: 8 Gy/1 fraction SBRT: 25 Gy/5 fractions	5.1% ^A^
Weiss GJ, et al. [60]	2017	I	11	Pembrolizumab	Gemcitabine (Gem)-based chemotherapy	18.2%
Kamath SD, et al. [61]	2020	I	21 ^B^	Arm A: Ipilimumab 3 mg/kg Arm B: Ipilimumab 3 mg/kg Arm C: Ipilimumab 6 mg/kg	Gem 750 mg/m^2^ Gem 1g/m^2^ Gem 1g/m^2^	14% ^C^

Abbreviation: PDAC: pancreatic ductal adenocarcinoma; ORR: objective response rate; SBRT: ^A^: The total ORR of four arms; ^B^: 67% of them received at least one line of chemotherapy; ^C^: The total ORR of three arms.

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
