# Peer review of "KRAS Mutation Dictates the Cancer Immune Environment in Pancreatic Ductal Adenocarcinoma and Other Adenocarcinomas"

_cancers, 2021, doi:10.3390/cancers13102429_

Round 1

Reviewer 1 Report

This review by Gu et al considers the role of RAS mutations in modulating the immune environment in different types of cancer.  It is a timely consideration, bearing in mind the state of the art, with multiple RAS inhibitors now reported in pre-clinical studies and K12C inhibitors now giving positive clinical results.    There are a couple of minor issues (listed below) but the main weaknesses are lack of depth in covering the molecular basis of the differences between the different cancer types and lack of consideration of recent exciting results using K12C inhibitors with Immune Checkpoint therapy (e.g. Canon et al).

minor issues;

  1. Ln 42: which BRCA gene.
  2. Ln 67: what is a massive myeloid cell

Author Response

Response to Reviewer 1 Comments

Point 1: There are a couple of minor issues (listed below) but the main weaknesses are lack of depth in covering the molecular basis of the differences between the different cancer types and lack of consideration of recent exciting results using K12C inhibitors with Immune Checkpoint therapy (e.g. Canon et al).

Response 1: Thank you for providing such professional comments on our manuscript. We believe the scientific quality of this manuscript will be improve if revising according to your suggestions. The following information are our responses to your comments. You can find the revised information, which have been labeled in yellow in the current form.

1.One of your major concerns is a lack of depth in covering the molecular basis of the differences between different cancer types. As you suggested, we have added information associated with KRAS mutation in causing tumoral immunosuppression in PDAC, CRAC and LUAC in Table 1, in order to avoid the length of the current text to be too long. Meanwhile, we have labeled the relevant information in the main text, which are associated with the molecular basis.

2.The other major concern is a lack of consideration of recent data associated with KRASG12C inhibition with the ICB therapy. As you suggested, we have added information associated with AMG510 and MRTX849 in treating KRASG12C-mutated LUAC and CRAC. You can find them in the last paragraph of the section 4.Current status of immune checkpoint blockade therapy for PDAC. This question is also raised by the other peer reviewer. This paragraph mainly introduces the anti-tumoral effect of KRASG12C of AMG510 and MRTX849, along with their effects on improving the immune milieus in CRAC models. We have cited the references by Canon et al. and other authors.

Point 2: Minor issues:

Ln 42: which BRCA gene.

Ln 67: what is a massive myeloid cell

Response 2:

  1. In line 55 of the present form, BRCA should be BRCA1 and BRCA2.
  2. In line 81-82 of the present form, myeloid cells include neutrophils, MDSCs and M2-like macrophages.

Reviewer 2 Report

In this review article, the authors have provided a concise summary of the current literature on the carcinogenic role of KRAS mutation and immune environment in pancreatic ductal adenocarcinoma (PDAC) and other adenocarcinomas; colorectal adenocarcinoma (CRAC) and lung adenocarcinoma (LUAC). This article explains how the immunosuppressive environment in PDAC makes monotherapy by using immune checkpoint blockade (ICB) drugs almost completely ineffective.  This article aims to uncover the mechanism by which KRAS-mutation dictates the cancer-immune status across these adenocarcinomas. However, it lacks a discussion on the future direction of the treatment strategies to overcome the KRAS-mutation-driven immunosuppression in PDAC. Moreover, the tables in the manuscript nicely summarize the recent literature to compare the immune-related characteristics among KRAS-mutant adenocarcinomas and the effectiveness of ICB therapy on PDAC. The note-chart (Figure 1) showing KRAS mutation-induced growth and immunosuppression events could be modified into a nice graphical abstract of KRAS-mutation-induced events in PDAC. Overall, this manuscript is very organized and well written.

Author Response

Response to Reviewer 2 Comments

Point 1: This article aims to uncover the mechanism by which KRAS-mutation dictates the cancer-immune status across these adenocarcinomas. However, it lacks a discussion on the future direction of the treatment strategies to overcome the KRAS-mutation-driven immunosuppression in PDAC. 

Response 1:

Thank you for providing such professional comments on our manuscript. According to your suggestions, we have revised this manuscript.

Your major concern is a discussion on the future direction of the treatment strategies to overcome the KRAS mutation-driven immunosuppression in PDAC. This suggestion is reasonable undoubtedly. We have added advances associated with KRASg12c inhibitors in the last paragraph of the section 4.Current status of immune checkpoint blockade therapy for PDAC. This question is also raised by the other peer reviewer. This paragraph mainly introduces the anti-tumoral effect of KRASg12c inhibitors, such as AMG510 and MRTX849, along with their effects on improving the immune milieus in CRAC models. Due to a lack of researches in PDAC, we point out the future direction of using KRASg12c inhibition plus anti-PD-1 therapy to overcome immunosuppression in PDAC. Moreover, due to majority of PDAC patients harbor G12D missense mutations in their tumors, another future direction in overcoming immunosuppression in PDAC relies on new drug development or other approach establishment in this field. You can find details in this paragraph. For your concern, I have labeled the information associated with the future direction to overcome immunosuppression in PDAC in blue.

Point 2: Moreover, the tables in the manuscript nicely summarize the recent literature to compare the immune-related characteristics among KRAS-mutant adenocarcinomas and the effectiveness of ICB therapy on PDAC. The note-chart (Figure 1) showing KRAS mutation-induced growth and immunosuppression events could be modified into a nice graphical abstract of KRAS-mutation-induced events in PDAC.

Response 2: We still believe the current figure 1 is more suitable here by using a note chart, because we had tried to design a graphical figure, but it looked like so ugly here. So, we want to use the note chart here.

Round 2

Reviewer 1 Report

The manuscript has been improved